# Transgenic Overexpression of HDAC9 Promotes Adipocyte Hypertrophy, Insulin Resistance and Hepatic Steatosis in Aging Mice

**DOI:** 10.3390/biom14040494

**Published:** 2024-04-18

**Authors:** Praneet Veerapaneni, Brandee Goo, Samah Ahmadieh, Hong Shi, David S. Kim, Mourad Ogbi, Stephen Cave, Ronnie Chouhaita, Nicole Cyriac, David J. Fulton, Alexander D. Verin, Weiqin Chen, Yun Lei, Xin-Yun Lu, Ha Won Kim, Neal L. Weintraub

**Affiliations:** 1Vascular Biology Center, Medical College of Georgia, Augusta University, 1460 Laney Walker Blvd, Augusta, GA 30912, USA; pveerapaneni@augusta.edu (P.V.); bgoo@augusta.edu (B.G.); sahmadieh@augusta.edu (S.A.); hoshi@augusta.edu (H.S.); skim1@augusta.edu (D.S.K.); mogbi@augusta.edu (M.O.); scave@augusta.edu (S.C.); rchouhaita@augusta.edu (R.C.); ncyriac@augusta.edu (N.C.); dfulton@augusta.edu (D.J.F.); averin@augusta.edu (A.D.V.); hkim3@augusta.edu (H.W.K.); 2Department of Medicine, Medical College of Georgia, Augusta University, 1460 Laney Walker Blvd, Augusta, GA 30912, USA; 3Department of Pharmacology and Toxicology, Medical College of Georgia, Augusta University, 1460 Laney Walker Blvd, Augusta, GA 30912, USA; 4Department of Physiology, Medical College of Georgia, Augusta University, 1460 Laney Walker Blvd, Augusta, GA 30912, USA; wechen@augusta.edu; 5Department of Neuroscience and Regenerative Medicine, Medical College of Georgia, Augusta University, 1460 Laney Walker Blvd, Augusta, GA 30912, USA; ylei@augusta.edu (Y.L.); xylu@augusta.edu (X.-Y.L.)

**Keywords:** HDAC9, overexpression, insulin resistance, adipose tissue, liver

## Abstract

Histone deacetylase (HDAC) 9 is a negative regulator of adipogenic differentiation, which is required for maintenance of healthy adipose tissues. We reported that *HDAC9* expression is upregulated in adipose tissues during obesity, in conjunction with impaired adipogenic differentiation, adipocyte hypertrophy, insulin resistance, and hepatic steatosis, all of which were alleviated by global genetic deletion of *Hdac9*. Here, we developed a novel transgenic (TG) mouse model to test whether overexpression of *Hdac9* is sufficient to induce adipocyte hypertrophy, insulin resistance, and hepatic steatosis in the absence of obesity. HDAC9 TG mice gained less body weight than wild-type (WT) mice when fed a standard laboratory diet for up to 40 weeks, which was attributed to reduced fat mass (primarily inguinal adipose tissue). There was no difference in insulin sensitivity or glucose tolerance in 18-week-old WT and HDAC9 TG mice; however, at 40 weeks of age, HDAC9 TG mice exhibited impaired insulin sensitivity and glucose intolerance. Tissue histology demonstrated adipocyte hypertrophy, along with reduced numbers of mature adipocytes and stromovascular cells, in the HDAC9 TG mouse adipose tissue. Moreover, increased lipids were detected in the livers of aging HDAC9 TG mice, as evaluated by oil red O staining. In conclusion, the experimental aging HDAC9 TG mice developed adipocyte hypertrophy, insulin resistance, and hepatic steatosis, independent of obesity. This novel mouse model may be useful in the investigation of the impact of *Hdac9* overexpression associated with metabolic and aging-related diseases.

## 1. Introduction

Histone deacetylases (HDACs) are epigenetic regulators that broadly orchestrate chromatin structure and gene expression. HDACs are categorized into five major classes based on their sequence homology. Amongst the 18 members of the HDAC family, HDAC9 is a class IIa HDAC that has weak enzymatic activity and is considered to function as a pseudoenzyme [1]. HDAC9 can shuttle freely between the nucleus and cytoplasm and regulate tissue-specific transcription by interacting with histone and non-histone substrates [2]. HDAC9 is involved in diverse physiological processes, including bone formation, adipocyte differentiation, and innate immunity [1]. Upregulated expression of *Hdac9* has been reported to contribute to the pathogenesis of various chronic diseases, including cardiovascular diseases (CVD), cancer, liver disease, and obesity [3,4,5].

The development and maintenance of healthy adipose tissues is fundamental to organismal homeostasis [6]. Although they are most widely recognized as a storage depot for excess calories, adipose tissues play a vital role in hormonal regulation, immune function, and heat generation. Adipose depots are broadly divided into visceral and subcutaneous based on anatomic location. The depots consist of mature, lipid-containing adipocytes and stromovascular cells, the latter of which include large numbers of partially-committed stem cells (i.e., preadipocytes). The differentiation of preadipocytes into adipocytes is a fundamental process required for adipose tissue formation, healthy remodeling, and expansion during obesity [7,8]. Impaired adipogenic differentiation is associated with adipocyte hypertrophy and may contribute to metabolic disease associated with both lipoatrophy and obesity, two apparently disparate conditions characterized by insulin resistance and ectopic accumulation of lipid in liver [9]. Thus, understanding the mechanisms that control adipose tissue biology may provide important insight into common metabolic diseases.

We previously reported that downregulation of HDAC9 is obligatory for the initiation of adipogenic differentiation in preadipocytes [10]. Additionally, we reported that *Hdac9* expression is upregulated in the adipose tissues of obese high fat diet-fed mice, in conjunction with impaired adipogenic differentiation, adipocyte hypertrophy, insulin resistance, and ectopic accumulation of lipid in the liver, all of which were ameliorated by global genetic deletion of *Hdac9* [11]. Further, Chen et al. reported that hepatic HDAC9 is involved in abnormal glucose metabolism and type 2 diabetes associated with viral hepatitis C infection [12]. Moreover, *Hdac9* was reported to be expressed in insulin-producing β-cells, and HDAC loss-of-function enhanced β-cell mass [13]. Thus, HDAC9 may have multiple functions pertinent to adipose biology, liver metabolism, and pancreatic function that in turn regulate key parameters of metabolic homeostasis.

Animal models of *Hdac9* gene deletion (i.e., global, tissue-specific knockout) and gene silencing approaches by viral transfection have been utilized to investigate the role of HDAC9 in vivo and in vitro. Such models are useful to study the impact of loss of expression of *Hdac9* in regulating cell and tissue function. Here, we created a novel HDAC9 transgenic (TG) mouse model in which *Hdac9* gene and HDAC9 protein are overexpressed globally to investigate the impact of increases in expression of *Hdac9*. This model was employed to test the hypothesis that overexpression of *Hdac9* in mice is sufficient to induce adipocyte hypertrophy, insulin resistance, and ectopic lipid accumulation in the liver, independent of diet-induced obesity.

## 2. Materials and Methods

### 2.1. HDAC9 TG Mice

HDAC9 TG mice were generated using the PiggyBac system by Cyagen Biosciences Inc. (Suzhou, China). A targeting vector was constructed using Vectorbuilder Inc. (Chicago, IL, USA). Briefly, the PiggyBac transposon vector comprised mouse *Hdac9* transgene preceded by CAG promoter and a Kozak translation initiation sequence with a polyadenylation stop region (rBG pA) and downstream ampicillin resistance sequence. The PiggyBac transposase underwent binding to transposon-specific inverted terminal repeats (3′ and 5′ ITRs) and inserted the ITR-expression cassette into the TTAA chromosomal site, and only a single copy was integrated into the genome per TTAA site, resulting in a more consistent expression of the transgene [14]. For pronuclear injection, the PiggyBAC transposon (with the *Hdac9* transgene) and transposase enzymes were co-injected into embryonic stem cells to generate transgenic mice. WT and HDAC9 TG mice were bred to obtain male HDAC9 TG and HDAC9 WT littermate controls for experiments. Mice were fed a standard laboratory (i.e., chow) diet after weaning. Body weights were measured weekly, along with food and water consumption. Ambulation was measured using a Comprehensive Laboratory Animal Monitoring System (CLAMS, Columbus Instruments, Columbus, OH, USA). All mice were euthanized under isoflurane anesthesia, after which tissues were collected, snap frozen in liquid nitrogen, and stored at −80 °C. All animal studies were conducted using a protocol approved by the Institutional Animal Care and Use Committee of the Medical College of Georgia at Augusta University.

### 2.2. Genotyping

Genomic DNA was extracted from mouse tails using a REDExtract N-AMP Kit (Sigma-Aldrich, Darmstadt, Germany), following the manufacturer’s protocol. Extracted DNA was amplified using primers listed in Appendix A, and PCR was performed using an Eppendorf Vapo.Protect Mastercycler^®^ Pro (Eppendorf, Hamburg, Germany) with 2×Taq Master Mix (Vazyme, Nanjing, China) via the following protocol: 94 °C for 3 min, 35 cycles of 94 °C for 1 min, 55 °C for 1 min, 72 °C for 1 min, and a final extension at 72 °C for 5 min. Detection of PCR products was performed using 1.5% agarose gel electrophoresis in 1XTAE buffer.

### 2.3. RT-PCR

RNA was extracted using an RNeasy Minikit (Qiagen GmbH, Hilden, Germany) according to the manufacturer’s instructions. RNA yield and purity were measured using a Nanodrop spectrophotometer. Quantification of mRNA levels was performed using SYBR Green qRT-PCR Kits (Agilent Technologies, Santa Clara, CA, USA). Fold change was calculated using the ΔΔCt method.

### 2.4. Western Blot

Protein was extracted from subcutaneous adipose tissues using a Tissue-Tearor (BioSpec, Bartlesville, OK, USA) in RIPA buffer with protease inhibitors, followed by centrifugation, separation by SDS-PAGE, transfer to nitrocellulose membrane, and probing with appropriate antibodies [HDAC9 (Biorbyt, Cambridge, UK) and Hsp90AA1 (Santa Cruz Biotechnology, Dallas, TX, USA)]. Blots were developed using an ECL system.

### 2.5. Plasma Insulin Measurement

Blood was collected with heparin and centrifuged at 4 °C and 2655× *g* for 20 min to isolate plasma. Plasma insulin was measured using an Abcam (Cambridge, UK) Mouse Insulin ELISA Kit (AB277390) according to the manufacturer’s instructions. 

### 2.6. Body Composition Measurements

Fat and lean mass were measured in one-year-old mice using nuclear magnetic resonance (NMR) spectroscopy (Bruker Minispec LF90II, Billerica, MA, USA) and normalized to total body weight as previously described [15].

### 2.7. Insulin Tolerance Test

At 18 weeks and 40 weeks of age, mice were fasted for 6 h, followed by intraperitoneal injection with regular insulin at 0.6 units/kg body weight. Glucose levels were measured at baseline and every 30 min up to 90 min following insulin injection as previously described [15].

### 2.8. Glucose Tolerance Test

At 18 weeks and 40 weeks of age, mice were fasted for 24 h, followed by intraperitoneal injection of glucose at 1 g/kg body weight. Glucose levels were measured via tail vein at baseline and every 30 min up to 2 h following glucose injection as previously described [15].

### 2.9. Histology

Frozen subcutaneous fat was transferred to 10% formalin, dehydrated in 70% ethanol, and embedded in paraffin. Subcutaneous tissue sections were stained with hematoxylin and eosin (H and E). Images were analyzed using ImageJ software (version 1.54i 03) to quantify adipocyte number and size using the Adiposoft plugin (Pamplona, Spain). Cells were traced by using the program’s freehand selection, and area was automatically calculated using the measure function as previously described [15]. Frozen liver was transferred to 30% sucrose and processed for OCT on a dry bed. Neutral lipids, stained with Oil red O (ORO), were visualized as previously described [16].

### 2.10. Noninvasive Blood Pressure Measurement

Blood pressure was measured using the tail-cuff method (CODA system, Kent Scientific Corporation, Torrington, CT, USA). Mice were trained for a period of three days to ensure quality and reproducibility of the data. Final systolic and diastolic values were obtained by averaging 20 individual measurements for each mouse.

### 2.11. Pulse Wave Velocity

Mice were anesthetized with isoflurane and placed on a pad equipped with echocardiogram capability. Here, 20 MHz probes were used to measure the ascending aortic and abdominal aortic flow. Velocity was measured by dividing distance between probes by the system-calculated ejection time using a high-frequency, high-resolution Doppler spectrum analyzer (DSPW, Indus Instruments, Houston, TX, USA).

### 2.12. Statistics

GraphPad Prism version 10.0.2 software (GraphPad Software, San Diego, CA, USA) was used for statistical calculations. Data are expressed as mean ± SEM (standard error of the mean). Differences between HDAC9 TG and WT controls were analyzed by unpaired two-tailed Student’s *t*-test. Welch’s correction was used for two-sample comparison between groups with unequal variances. Datasets with repeated time point measurements were analyzed using two-way ANOVA. *p* < 0.05 was considered statistically significant.

## 3. Results

### 3.1. Generation and Validation of HDAC9 TG Mice

HDAC9 TG mice were generated in WT C57BL/6 mice using the vector shown in Figure 1A. The genotypes of the mice were confirmed via gel electrophoresis, with HDAC9 TG mice showing bands at 352 bp and 511 bp, which were not detected in WT mice (Figure 1B and Appendix A). Overexpression of *Hdac9* in the mice was confirmed by RT-PCR in tails (Figure 1C) and subcutaneous (SQ) adipose tissues (Figure 1D). The generation of HDAC9 TG mice was further validated through HDAC9 protein expression in SQ adipose tissue using western blot (Figure 1E).

### 3.2. General Features of HDAC9 TG Mice

The HDAC9 TG mice appeared slightly smaller than their WT littermates (Figure 2A), but weaned at the expected age and were fertile. When housed at ambient temperature and fed a standard laboratory diet, the HDAC9 TG mice gained significantly less weight compared to the WT controls (Figure 2B,C). We performed body composition analysis using NMR spectroscopy at one year of age. There was a trend toward reduced whole body (Figure 2D) and fat mass (Figure 2E) and increased lean mass (Figure 2F) in the HDAC9 TG mice, though the differences were not statistically significant. Metabolic testing using CLAMS indicated that the HDAC9 TG mice consumed similar amounts of water and food compared to the WT mice (Figure 2G,H). The HDAC9 TG mice showed a strong trend toward increased ambulation compared to WT mice, though the differences were not statistically significant (Figure 2I,J).

HDAC9 has been implicated in stroke and cardiovascular disease [17]. Therefore, we assessed blood pressure and pulse wave velocity non-invasively in 10-month-old HDAC9 TG and WT control mice. We detected no significant differences in systolic pressure (Appendix A), diastolic pressure (Appendix A), or pulse wave velocity (Appendix A) between the two genotypes.

### 3.3. Impaired Insulin Sensitivity and Glucose Tolerance in Aging HDAC9 TG Mice

We previously reported that in obesity, HDAC9 expression is increased in the adipose tissues of humans and mice, in conjunction with impaired insulin sensitivity and glucose tolerance [15]. Moreover, global deletion of *Hdac9* improved insulin sensitivity and glucose tolerance in both male and female mice [11]. Therefore, we assessed insulin and glucose tolerance in the WT and HDAC9 TG mice at different ages. When tested at a young age (18 weeks), we detected no differences in glucose tolerance (Figure 3A) or insulin sensitivity (Figure 3B) between the two genotypes. However, when the mice were tested at middle age (40 weeks), both insulin sensitivity (Figure 3C) and glucose tolerance (Figure 3D) were impaired in the HDAC9 TG mice.

### 3.4. Reduced Inguinal Adipose Tissue in HDAC9 TG Mice

At the time of sacrifice, we harvested adipose tissue depots and various internal organs from one-year-old WT and HDAC9 TG mice. Notably, the HDAC9 TG mice had smaller total adipose tissue mass compared to WT mice (Figure 4A), which was attributed principally to reduced inguinal subcutaneous fat (Figure 4B); there were no significant differences in epididymal adipose tissue (Figure 4C) or brown adipose tissue (BAT, Figure 4D) between the two genotypes. Moreover, there were no significant differences in heart (Figure 4E), liver (Figure 4F), kidney (Figure 4G), spleen (Figure 4H), gastrocnemius muscle (Figure 4I), or testis (Figure 4J) weight in HDAC9 TG versus WT mice.

### 3.5. Morphological Features of Adipose Tissue, Liver and Skeletal Muscle in HDAC9 TG Mice

Subcutaneous adipose tissues from 10-month-old WT and HDAC9 TG mice were processed for histology (H and E staining). Interestingly, the HDAC9 TG mice exhibited reduced numbers of mature adipocytes (Figure 5A,B), and increased mean adipocyte size (Figure 5A,C) without alterations in adipogenic gene expression (Appendix A). Also, fewer stromovascular cells were detected in the subcutaneous adipose tissues of the HDAC9 TG mice compared to the WT mice (Figure 5A,D). We detected no differences in plasma cholesterol (Appendix A) and random (non-fasting) insulin levels in the HDAC9 TG mice compared to WT mice (Appendix A). Interestingly, there was significantly more lipid accumulation (detected via ORO staining, Figure 5E,F) along with reduced expression of carnitine-acylcarnitine translocase (*Cact*), which is associated with fatty acid oxidation [18] (Appendix A), in the livers of HDAC9 TG mice compared to the WT mice. There was no significant difference in skeletal muscle morphology (Figure 5G) between WT and HDAC9 TG mice, and no lipid was detected in the skeletal muscles of the HDAC9 TG mice via ORO staining (Appendix A).

## 4. Discussion

We previously reported that *Hdac9* expression in adipose tissue is increased in obese high fat fed-mice, in conjunction with adipocyte hypertrophy, ectopic lipid accumulation in the liver, insulin resistance, and glucose intolerance [11]. Global genetic deletion of *Hdac9* resulted in reduced weight gain, improved glucose tolerance and insulin sensitivity, and reduced ectopic lipid accumulation in high fat fed-mice, suggesting that HDAC9 plays a key role in obesity-related metabolic disease [11]. Here, we tested whether overexpression of *Hdac99* is sufficient to produce features of metabolic disease in the absence of obesity. While young (18-week-old) HDAC9 TG mice appeared metabolically healthy, middle aged (40-week-old) HDAC9 TG mice exhibited impaired glucose tolerance and insulin resistance, along with increased lipid accumulation in the liver. These features of metabolic disease occurred in mice fed a standard laboratory diet, suggesting that *Hdac9* overexpression is sufficient to induce metabolic disease in the context of aging, independent of obesity.

HDAC9 is a class 2a histone deacetylase that is expressed in many tissues, including the brain, heart, hematopoietic cells, skeletal muscle, and fat. Multiple studies have concluded that HDAC9 contributes to a variety of chronic diseases, including cardiovascular diseases, cancer, and obesity [3,4,5]. In adipose tissues, *Hdac9* expression is highest in the stromovascular fraction, and in isolated preadipocytes, downregulation of HDAC9 is obligatory for the adipogenic differentiation program to ensue. Thus, HDAC9 functions as a negative regulator of adipogenic differentiation [10]. Like other class 2a HDACs, HDAC9 exhibits weak deacetylase activity, and its enzymatic domain is dispensable for regulation of adipogenic differentiation [10]. Interestingly, we observed impaired adipogenic differentiation, and increased *Hdac9* expression, in inguinal adipose tissues of mice fed a high fat diet and housed at thermoneutral temperatures [19]. Under these conditions, deletion of *Hdac9* resulted in increased energy expenditure and heat generation, in conjunction with browning of inguinal adipose tissues, which likely contributed to the favorable metabolic phenotype. In contrast, housing mice at ambient temperature while feeding a high fat diet resulted in improved metabolic health, along with minimal changes in adipose *Hdac9* expression and adipogenic differentiation. Short-term studies have indicated that inguinal adipose *Hdac9* expression is sensitive to changes in housing temperature via a TRPM8-dependent mechanism [19]. Thus, HDAC9 may function as a metabolic switch in adipose tissues, regulating the balance between energy storage and heat production to maintain homeostasis depending on environmental conditions [19].

In order to test whether overexpression of HDAC9 is sufficient to promote adipose tissue dysfunction and insulin resistance, we developed a global HDAC9 transgenic mouse model. Morphologically, the HDAC9 TG mice were slightly smaller than their WT littermates, but developed normally and exhibited no obvious aberrant behaviors. When monitored up to 40 weeks of age, the HDAC9 TG mice exhibited no increase in mortality. The gross structure of their vital organs, including the heart, kidneys, and liver, was unremarkable, and the organ weights were similar to their WT counterparts. The testes were also similar in appearance and weight between the wild-type and TG mice. Interestingly, there was a trend towards increased spleen mass in the HDAC9 TG mice (statistically non-significant). *Hdac9* is expressed in hematopoietic cells, and deletion of *Hdac9* in the bone marrow protected hypercholesterolemic mice against the development of atherosclerosis [20]. Here, we assessed blood pressure and pulse wave velocity (a parameter of aortic stiffness) and did not detect differences between WT and HDAC9 TG mice (Appendix A). Nevertheless, more extensive investigations will be required to determine whether overexpression of *Hdac9* promotes cardiovascular disease under different experimental conditions.

Body composition analysis showed a trend toward reduced fat mass in the HDAC9 transgenic mice, and quantification of adipose tissue depots confirmed that inguinal subcutaneous fat, in particular, was diminished in the HDAC9 TG mice. Histologic examination of the subcutaneous adipose tissues demonstrated evidence of reduced numbers of mature adipocytes, along with adipocyte hypertrophy, without altering adipogenic gene expression, in aging HDAC9 TG mice. Interestingly, the number of stromovascular cells (which are enriched in partially differentiated stem cells, i.e., preadipocytes) was significantly reduced in the adipose tissues of HDAC9 TG mice compared to WT mice. Preadipocytes play a key role in maintaining adipose tissue health by differentiating into mature adipocytes, which are required to replenish those that turn over during healthy aging and to expand adipose depots during chronic caloric excess [21,22]. This raises the possibility that depletion of the preadipocyte pool may, over time, have resulted in diminished numbers of mature adipocytes, thus promoting adipocyte hypertrophy and lipid deposition in the liver. Adipocyte hypertrophy is associated with cellular stress, inflammation, fibrosis, and reduced insulin sensitivity [23,24], suggesting that this mechanism may have contributed to the adverse metabolic phenotype observed in the aging HDAC9 TG mice. However, additional studies will be required to test this hypothesis.

Over the course of this study, we tested insulin sensitivity and glucose tolerance in both young (18-week-old) and aging (40-week-old) mice. At 18 weeks of age, the HDAC9 TG mice exhibited normal glucose tolerance and insulin sensitivity, suggesting that insulin production and bioactivity were preserved at this time. However, we did not systematically study pancreas function in our mice. *Hdac9* is expressed in insulin producing β-cells [13], and mice globally deficient in *Hdac9* exhibited an increase in β-cell mass [11]. Thus, it is possible that over the course of this study, overexpression of *Hdac9* could have led to β cell dysfunction/depletion and reduced insulin secretion, which in turn could have contributed to the adverse metabolic phenotype (i.e., adipocyte hypertrophy, hepatic steatosis) observed in the aged HDAC9 TG mice. Further studies will be needed to test this hypothesis.

It is important to point out that we only investigated one line of HDAC9 TG mice. However, our findings with these mice were generally consistent with what we hypothesized based on data we observed in obese mice (where endogenous *Hdac9* is upregulated), and opposite to what we observed in *Hdac9* knockout mice [11]. This increases the confidence that the phenotype of the mice was, in fact, due to *Hdac9* overexpression. Additionally, we only studied male mice; we recently reported sexual dimorphism with regard to the impact of adipocyte-specific *Hdac9* gene deletion on high fat diet-induced obesity, wherein only the female adipose-specific *Hdac9* knockout mice exhibited a strong phenotype [15]. Additionally, our study was conducted in an ambient temperature environment. It is possible that *Hdac9* overexpression could have a lesser impact under thermoneutral conditions, where endogenous *Hdac9* expression in adipose tissues is upregulated [19]. Finally, in this study, mice were fed a standard laboratory diet and were therefore lean, so our findings cannot be extrapolated to murine models of obesity.

## 5. Conclusions

We report that global transgenic overexpression of *Hdac9* is sufficient to promote adipocyte hypertrophy, hepatic steatosis, insulin resistance, and glucose intolerance in aging mice fed a standard laboratory diet. While the mechanisms remain to be determined, we detected reduced numbers of mature adipocytes, along with reduced numbers of stromovascular cells, in the adipose tissues of the aging HDAC9 TG mice. This suggests that the HDAC9 TG mice may not have been able to efficiently replenish adipocytes that turn over during healthy aging, thus contributing to lipid deposition in the liver and reduced insulin sensitivity. These findings may have implications for aging-related metabolic disease.

## Figures and Tables

**Figure 1 biomolecules-14-00494-f001:**
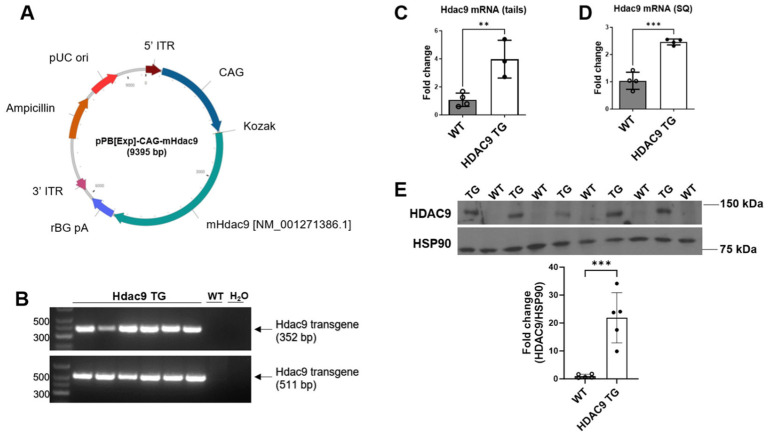
Generation and validation of HDAC9 TG mice. (**A**) Mapping the transgene. The targeting vector (PiggyBac transposon gene expression vector) comprised mouse *Hdac9* transgene preceded by CAG promoter and a Kozak translation initiation sequence, with a polyadenylation stop region (rBG pA) and downstream ampicillin resistance sequence. For pronuclear injection, the PiggyBAC transposon (with the *Hdac9* transgene) and transposase enzymes were co-injected into embryonic stem cells. (**B**) Genotyping results. (**C**) *Hdac9* mRNA expression in tails. (**D**) *Hdac9* mRNA expression in SQ adipose tissue. (**E**) HDAC9 protein expression in SQ adipose tissue (HSP90 was used as a loading control). Original Western blot images are contained in Appendix A. Data are expressed as mean ± SEM (*n* = 3–4). ** *p* < 0.01, *** *p* < 0.001.

**Figure 2 biomolecules-14-00494-f002:**
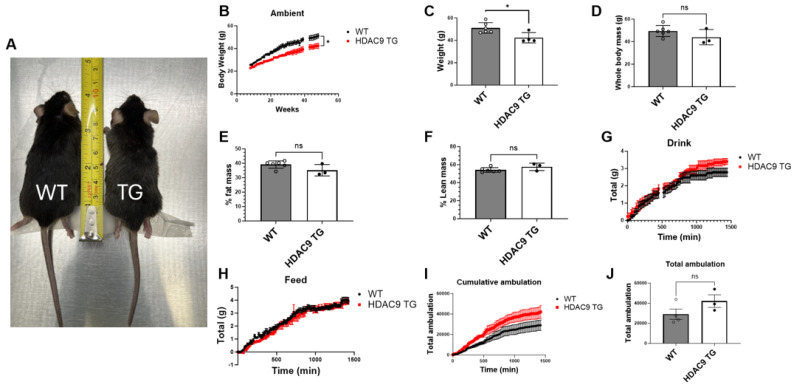
General features of HDAC9 TG mice. Representative picture of 40-week-old male WT and HDAC9 TG mice (**A**). Growth curves of mice (**B**) and average body weight after 37 weeks under ambient temperature (**C**). Whole body mass (**D**), % fat mass (**E**), and % lean mass (**F**) as measured by NMR. Water consumption (**G**), food consumption (**H**), cumulative ambulation (**I**), and total ambulation (**J**) were measured by CLAMS. Data are expressed as mean ± SEM (*n* = 3–6). * *p* < 0.05, ns = not significant.

**Figure 3 biomolecules-14-00494-f003:**
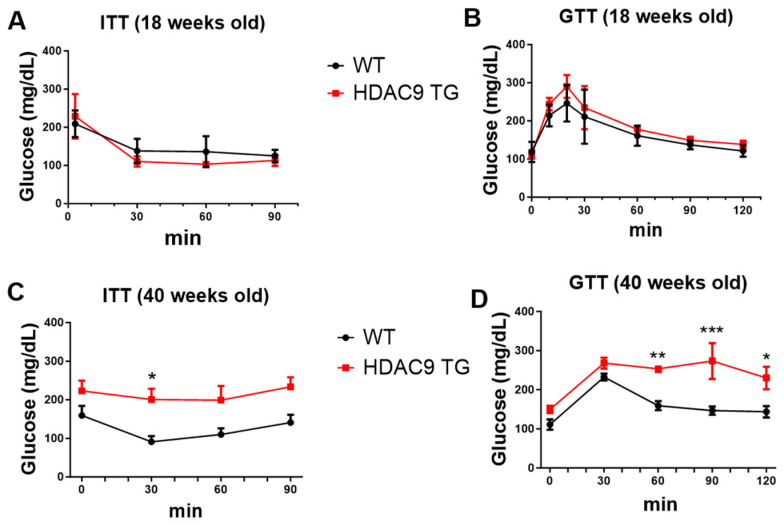
Impaired insulin sensitivity and glucose tolerance in aging male HDAC9 TG mice. ITT and GTT in 18-week-old (**A**,**B**) versus 40-week-old (**C**,**D**) mice. Data are expressed as mean ± SEM (*n* = 3–4). * *p* < 0.05, ** *p* < 0.01, *** *p* < 0.001.

**Figure 4 biomolecules-14-00494-f004:**
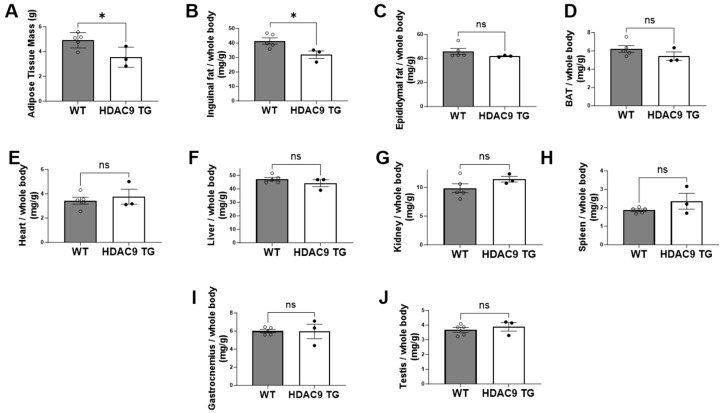
Reduced weight of inguinal adipose tissues in HDAC9 TG mice as compared to WT mice (40 weeks old, male). Total adipose tissue mass (**A**) or individual tissue weight normalized to whole body weight (**B**), inguinal fat; (**C**), epididymal fat; (**D**), BAT; (**E**), heart; (**F**), liver; (**G**), kidney; (**H**), spleen; (**I**), gastrocnemius; (**J**), testis). Data are expressed as mean ± SEM (*n* = 3–5). * *p* < 0.05, ns = not significant.

**Figure 5 biomolecules-14-00494-f005:**
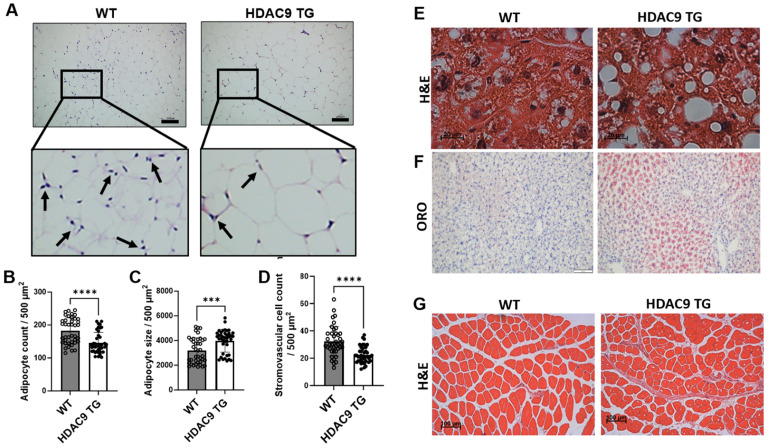
Morphological features of adipose tissue, liver, and skeletal muscle of aging HDAC9 TG mice (40 weeks old, male). (**A**) Representative images of inguinal adipose tissues. Arrows indicate stromovascular cells. Scale bar = 100 µm. (**B**) Quantification of adipocyte count. (**C**) Quantification of adipocyte size. (**D**) Quantification of stromovascular cell numbers. (**E**) Representative H and E staining images of liver. Scale bar = 20 µm. (**F**) Representative ORO staining images of liver. Scale bar = 100 µm. (**G**) Representative H and E staining images of skeletal muscle. Data are expressed as mean ± SEM (*n* = 40). Scale bar = 100 µm. *** *p* < 0.001, **** *p* < 0.0001.

## Data Availability

The data presented in this study are available within the article or Appendix A.

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
