# Peer review of "Transgenic Overexpression of HDAC9 Promotes Adipocyte Hypertrophy, Insulin Resistance and Hepatic Steatosis in Aging Mice"

_biomolecules, 2024, doi:10.3390/biom14040494_

Round 1
Reviewer 1 Report
Comments and Suggestions for Authors
Although the findings that the authors developed a novel transgenic (TG) mouse model to test whether overexpression of histone deacetylase 9 (HDAC9) was sufficient to induce adipocyte hypertrophy, insulin resistance and hepatic steatosis in the absence of obesity are interesting, numbers of points need clarifying and certain statements require further justification. These are given below.
<Point>
1. The authors made Hdac9 transgenic mice and analyzed in a line of them. Transgene is affected by the position of integration as shown in Figure 1E. The authors should examine at least two independent lines.
2. In animal experiments, the authors’ experiments should be approved by the so-called animal experiment committee (Animal Care and Use Committee of Medical College of Georgia at Augusta University) and described the approval number(s) and date. The authors described, “The animal study protocol was approved by the Institutional Review Board of Augusta University (Protocol code 2013-0528 and approved May 28, 2013)”. The approval is over 10 years old. It is assumed that there have been subsequent changes in regulations and laws, but were they effective at the time the research was carried out?
3. In Figure 2A, scale bar should be added because figures are usually magnified and/or reduced by printer.
4. In Figure 1 legend, the authors described, “HDAC9 TG mice were generated in WT C57BL/6 mice using the strategy shown in Figure 1A.” However, Figure 1A did not show the strategy but a construct for generation of the transgenic mice as described in the legend.
5. Concerning Figure 1B, the authors did not show primer position and or sequence. Therefore, why the 352 bp and 511 bp bands were generated only in HDAC9 TG was not clarified.
6. “Agilent Technologies, Santa Clara, USA” (line 108) should be changed to “Agilent Technologies, Santa Clara, CA”.
7. “BioSpec, Bartlesville, USA” (lines 112-113) should be changed to “BioSpec, Bartlesville, OK”.
8. “Santa Cruz Biotechnology, Dallas, USA)” (line 116) should be changed to “Santa Cruz Biotechnology, Dallas, TX”.
9. “Bruker Minispec LF90II, Billerica, USA” (line 125) should be changed to “Bruker Minispec LF90II, Billerica, MA”.
10. “Torrington, USA” (line 151) should be changed to “Torrington, CT”.
11. “Houston, USA” (line 160) should be changed to “Houston, TX”.
12. “San Diego, USA” (line 163) should be changed to “San Diego, CA”.
13. In Ref. 2, “Life (Basel) 2021, 11” should be changed to “Life (Basel) 2021, 11, 90”.
14. In Ref. 12, “Obesity (Silver Spring) 2023” should be changed to “Obesity (Silver Spring) 2023, 32, 107-119”.
15. In Ref. 15, “Chan, P.-C. The Role of Adipocyte Hypertrophy and Hypoxia in the Development of Obesity-Associated Adipose Tissue Inflammation and Insulin Resistance; IntechOpen: sine loco, 2017” should be changed to “Chen, P.-C.; Hsieh, P.-S. The Role of Adipocyte Hypertrophy and Hypoxia in the Development of Obesity-Associated Adipose Tissue Inflammation and Insulin Resistance in Adiposity–Omics and molecular understanding, Gordeladze, J. O. Ed., InTech, Rijeka, Croatia, 2017”.
Author Response
Response to Reviewers
We appreciate the Reviewers’ critical review and constructive suggestions for this manuscript. In response to their comments, we have revised the manuscript by re-organizing data, correcting errors in statistics and mouse generation strategy, and adding new data and supportive materials. We believe that the revised manuscript is substantially improved by these revisions. Following is a detailed, point-by-point responses to the Reviewers.
Reviewer 1
Although the findings that the authors developed a novel transgenic (TG) mouse model to test whether overexpression of histone deacetylase 9 (HDAC9) was sufficient to induce adipocyte hypertrophy, insulin resistance and hepatic steatosis in the absence of obesity are interesting, numbers of points need clarifying and certain statements require further justification. These are given below.
- The authors made Hdac9transgenic mice and analyzed in a line of them. Transgene is affected by the position of integration as shown in Figure 1E. The authors should examine at least two independent lines.
Authors' response: We thank the Reviewer for this valuable comment. We agree that expression levels of HDAC9 transgene could be affected by the integration position in the genome. To achieve more consistent transgene expression, PiggyBAC system was used, in which transposons integrate a copy of the transgene into the flanking TTAA sequence, leading to more precise and consistent expression of the target gene. Note that the transgenic mouse line was constructed via a contract with Cyagen Inc. We received only one line of HDAC9 TG mice from Cyagen, so unfortunately, we cannot study another line. However, our findings with these mice were generally consistent with what we hypothesized based on data we observed in obese mice (where endogenous HDAC9 is upregulated), and opposite to what we observed in Hdac9 knockout mice. This increases the confidence that the phenotype of the mice was, in fact, due to Hdac9 overexpression. In the revised manuscript, we have modified Materials and Methods and Discussion sections accordingly.
- In animal experiments, the authors’ experiments should be approved by the so-called animal experiment committee (Animal Care and Use Committee of Medical College of Georgia at Augusta University) and described the approval number(s) and date. The authors described, “The animal study protocol was approved by the Institutional Review Board of Augusta University (Protocol code 2013-0528 and approved May 28, 2013)”. The approval is over 10 years old. It is assumed that there have been subsequent changes in regulations and laws, but were they effective at the time the research was carried out?
Authors' response: We apologize for lack of clarity. This animal protocol (2013-0528) was first approved in 2013, and formally renewed triennially. In addition, our animal protocol has been reviewed and approved on an as- needed basis by the University’s Review Board. The final approval year of our animal protocol is 2023. To avoid confusion, we have modified the sentence in Institutional Review Board Statement section.
- In Figure 2A, scale bar should be added because figures are usually magnified and/or reduced by printer.
Authors' response: We took a new picture with a ruler included to show the size of mice (Figure 2A).
- In Figure 1 legend, the authors described, “HDAC9 TG mice were generated in WT C57BL/6 mice using the strategy shown in Figure 1A.” However, Figure 1A did not show the strategy but a construct for generation of the transgenic mice as described in the legend.
Authors' response: Thank you for pointing this out and we apologize for the mistake. We have modified the description for HDAC9 TG generation in Materials and Methods, results and legends of Figure 1.
- Concerning Figure 1B, the authors did not show primer position and or sequence. Therefore, why the 352 bp and 511 bp bands were generated only in HDAC9 TG was not clarified.
Authors' response: We apologize for lack of clarity. In the revised manuscript, we have included PCR and primers information in Supplementary Materials.
- “Agilent Technologies, Santa Clara, USA” (line 108) should be changed to “Agilent Technologies, Santa Clara, CA”.
Authors' response: Corrected.
- “BioSpec, Bartlesville, USA” (lines 112-113) should be changed to “BioSpec, Bartlesville, OK”.
Authors' response: Corrected.
- “Santa Cruz Biotechnology, Dallas, USA)” (line 116) should be changed to “Santa Cruz Biotechnology, Dallas, TX”.
Authors' response: Corrected.
- “Bruker Minispec LF90II, Billerica, USA” (line 125) should be changed to “Bruker Minispec LF90II, Billerica, MA”.
Authors' response: Corrected.
- “Torrington, USA” (line 151) should be changed to “Torrington, CT”.
Authors' response: Corrected.
- “Houston, USA” (line 160) should be changed to “Houston, TX”.
Authors' response: Corrected.
- “San Diego, USA” (line 163) should be changed to “San Diego, CA”.
Authors' response: Corrected.
- In Ref. 2, “Life (Basel) 2021, 11” should be changed to “Life (Basel) 2021, 11, 90”.
Authors' response: Corrected.
- In Ref. 12, “Obesity (Silver Spring) 2023” should be changed to “Obesity (Silver Spring) 2023, 32, 107-119”.
Authors' response: Corrected.
- In Ref. 15, “Chan, P.-C. The Role of Adipocyte Hypertrophy and Hypoxia in the Development of Obesity-Associated Adipose Tissue Inflammation and Insulin Resistance; IntechOpen: sine loco, 2017” should be changed to “Chen, P.-C.; Hsieh, P.-S. The Role of Adipocyte Hypertrophy and Hypoxia in the Development of Obesity-Associated Adipose Tissue Inflammation and Insulin Resistance in Adiposity–Omics and molecular understanding, Gordeladze, J. O. Ed., InTech, Rijeka, Croatia, 2017”.
Authors' response: Corrected.
Reviewer 2 Report
Comments and Suggestions for Authors
In this work, the authors report the results obtained by a novel transgenic mouse model, HDAC9 TG mice, developed to test whether overexpression of HDAC9 is sufficient to induce adipocyte hypertrophy, insulin resistance and hepatic steatosis in the absence of obesity. They reported that HDAC9 TG mice gained less body weight than wild-type (WT) mice when fed a standard laboratory diet for up to 40 weeks, which was attributed to reduced fat mass (primarily inguinal adipose tissue). At 40 weeks of age, HDAC9 TG mice exhibited impaired insulin sensitivity and glucose intolerance. Tissue histology demonstrated adipocyte hypertrophy, along with reduced numbers of mature adipocytes and stromovascular cells in the HDAC9 TG mouse adipose tissue. Oil red O staining revealed increased lipid in the liver of aging HDAC9 TG mice. Their conclusion is that aging HDAC9 TG mice develop adipocyte hypertrophy, insulin resistance, and hepatic steatosis, independent of obesity.
The work is potentially interesting, but some points needed to be improved and better investigate. In present form, I do not recommend its publication in Biomolecules journal.
Below are the comments:
1) Captions of each figure should be bigger.
2) Sex, numbers and initial age of mice used in the experiments should be inserted.
3) In Figure 1, a molecular weight marker should be included in both gel electrophoresis (mRNA and protein gels). In figure 1B it has been loaded, but it is not visible enough.
4) In the legend of each figure, they should indicate if data are expressed as mean ± SEM and the number of mice used.
5) In Figure 3, authors compare insulin and glucose tolerance in both male and female mice of young (18 weeks) and middle age (40 weeks) mice. One question is: How these parameters are in wild type mice at 40 weeks of age? In my opinion, they should compare ITT and GTT in both wild type and HDAC9 TG mice at different ages (WT 40 wks versus WT 18 wks; HDAC9 TG 40 wks versus HDAC9 TG 18 wks). Moreover, in the same animals the also should evaluate other metabolic parameters such as the content of triglycerides, total cholesterol, HDL cholesterol and LDL cholesterol
6) In Figure 4, I would suggest to the authors to compare adipose tissue depots and the weight of various internal organs in both wild type and HDAC9 TG mice at different ages (WT 40 wks versus WT 18 wks, HDAC9 TG 40 wks versus HDAC9 TG 18 wks). These experiments should help authors to understand the influence of age on total adipose tissue or individual tissue weight.
7) How is the expression of HDAC9 in wild type mice? It is influenced by age? Authors should clarify this point.
8) In figure 5 they show that HDAC9 TG mice exhibits reduced numbers of mature adipocytes and increased mean adipocyte size without alterations of adipogenic gene expression. In my opinion, the expression of these genes should be investigated not in adipose tissue but in adipocytes derived from this tissue and put in culture.
9) They sustain that in the livers of HDAC9 TG mice there was significantly more lipid accumulation compared to the WT mice. To reinforce this data, the authors should also investigate the expression of some genes involved in fatty acid oxidation such as ACOX1, CACT, ACSL1 and CPT1alpha.
10) It is not clear how they measure adipocyte size; this point needed to be better clarified in Materials and Methods section
Comments on the Quality of English LanguageModerate editing is required
Author Response
Response to Reviewers
We appreciate the Reviewers’ critical review and constructive suggestions for this manuscript. In response to their comments, we have revised the manuscript by re-organizing data, correcting errors in statistics and mouse generation strategy, and adding new data and supportive materials. We believe that the revised manuscript is substantially improved by these revisions. Following is a detailed, point-by-point responses to the Reviewers.
Reviewer 2
In this work, the authors report the results obtained by a novel transgenic mouse model, HDAC9 TG mice, developed to test whether overexpression of HDAC9 is sufficient to induce adipocyte hypertrophy, insulin resistance and hepatic steatosis in the absence of obesity. They reported that HDAC9 TG mice gained less body weight than wild-type (WT) mice when fed a standard laboratory diet for up to 40 weeks, which was attributed to reduced fat mass (primarily inguinal adipose tissue). At 40 weeks of age, HDAC9 TG mice exhibited impaired insulin sensitivity and glucose intolerance. Tissue histology demonstrated adipocyte hypertrophy, along with reduced numbers of mature adipocytes and stromovascular cells in the HDAC9 TG mouse adipose tissue. Oil red O staining revealed increased lipid in the liver of aging HDAC9 TG mice. Their conclusion is that aging HDAC9 TG mice develop adipocyte hypertrophy, insulin resistance, and hepatic steatosis, independent of obesity.
The work is potentially interesting, but some points needed to be improved and better investigate. In present form, I do not recommend its publication in Biomolecules journal.
Below are the comments:
1) Captions of each figure should be bigger.
Authors' response: Thank you for the suggestions. Font size has been enlarged in all figures.
2) Sex, numbers and initial age of mice used in the experiments should be inserted.
Authors' response: We have included this information in the figure legends in the revised manuscript.
3) In Figure 1, a molecular weight marker should be included in both gel electrophoresis (mRNA and protein gels). In figure 1B it has been loaded, but it is not visible enough.
Authors' response: The original uncropped images (including a molecular weight marker) of Figure 1E have been included in Supplemental Materials. We have also replaced Figure 1B with a better quality image and include a molecular marker.
4) In the legend of each figure, they should indicate if data are expressed as mean ± SEM and the number of mice used.
Authors' response: We have included this information in the figure legends in the revised manuscript.
5) In Figure 3, authors compare insulin and glucose tolerance in both male and female mice of young (18 weeks) and middle age (40 weeks) mice. One question is: How these parameters are in wild type mice at 40 weeks of age? In my opinion, they should compare ITT and GTT in both wild type and HDAC9 TG mice at different ages (WT 40 wks versus WT 18 wks; HDAC9 TG 40 wks versus HDAC9 TG 18 wks). Moreover, in the same animals the also should evaluate other metabolic parameters such as the content of triglycerides, total cholesterol, HDL cholesterol and LDL cholesterol.
Authors' response: We apologize for the confusion. In Figure 3, we compared the mice at different ages (18 vs 40 weeks), but not different sexes (male vs female). As you can see, insulin sensitivity and glucose tolerance were impaired only in aging (40 weeks) HDAC9 TG mice. Insulin and glucose tolerance were similar in 18 and 40 weeks WT mice.
According to the Reviewer’s suggestion, we measured plasma cholesterol levels but did not observe the differences between WT and TG mice (additional lipid data were not available). This new data has been included in Supplementary Figure S4A in the revised manuscript.
6) In Figure 4, I would suggest to the authors to compare adipose tissue depots and the weight of various internal organs in both wild type and HDAC9 TG mice at different ages (WT 40 wks versus WT 18 wks, HDAC9 TG 40 wks versus HDAC9 TG 18 wks). These experiments should help authors to understand the influence of age on total adipose tissue or individual tissue weight.
Authors' response: Thank you for the suggestion. In this study, we focused on insulin resistance in aging TG mice, and thus, collected the organs only from those aging mice because young mice exhibited no differences in insulin sensitivity or glucose tolerance. We are currently investigating the association of HDAC9 and adipose tissue aging in a separate study (please see the response to question #7 below).
7) How is the expression of HDAC9 in wild type mice? It is influenced by age? Authors should clarify this point.
Authors' response: We appreciate the Reviewer for the insightful comments. We are currently investigating the role of HDAC9 in adipose tissue aging, but defining its role and mechanisms in senescence requires a significant amount time and resources, and we respectfully contend that these experiments are beyond the scope of the present manuscript. We hope to report this information in a separate paper in the future.
8) In figure 5 they show that HDAC9 TG mice exhibits reduced numbers of mature adipocytes and increased mean adipocyte size without alterations of adipogenic gene expression. In my opinion, the expression of these genes should be investigated not in adipose tissue but in adipocytes derived from this tissue and put in culture.
Authors' response: Thank you for the suggestions. We have very preliminary results (n=2) that adipogenic gene expression was not different in preadipocytes isolated from WT and HDAC9 TG mice. Unfortunately, 40 weeks old TG mice are currently not available, and it will take a considerable amount of time to produce aged mice from which additional preadipocytes could be derived for cell culture testing. As this would substantially delay publication of our manuscript, we respectfully believe such experiments are beyond the scope of this study.
9) They sustain that in the livers of HDAC9 TG mice there was significantly more lipid accumulation compared to the WT mice. To reinforce this data, the authors should also investigate the expression of some genes involved in fatty acid oxidation such as ACOX1, CACT, ACSL1 and CPT1alpha.
Authors' response: We appreciate the Reviewer’s excellent suggestion. We measured genes related to fatty acid oxidation. Interestingly, we found Cact expression was significantly reduced in the liver of HDAC9 TG mice, while there was no difference in Cpt1α. Cact plays an important role in breaking down fats, and reduced expression of the gene in HDAC9 TG mice could be one of the potential mechanisms of lipid accumulation in the liver of HDAC9 TG mice. This new result has been included in Results and Supplemental Materials sections in the revised manuscript.
10) It is not clear how they measure adipocyte size; this point needed to be better clarified in Materials and Methods section.
Authors' response: We quantified adipocyte count and size using Adiposoft plugin (Pamplona, Spain) and ImageJ software (version 1.54i 03) per instructions given by the manufacturers. Briefly, cells were traced by using the program's freehand selection and area was automatically calculated by program's measure function. We have modified the description in Materials and Methods section.
Reviewer 3 Report
Comments and Suggestions for Authors
The authors characterized a novel TG mouse and show that OE of HADC9 induces adipocyte hypertrophy, but also leads to impaired insulin resistance and hepatic steatosis. If well the authors indicate the limitations of their work, they also recognize that additional studies will be required to link HDAC9 with other metabolic diseases.
On the other hand, from my point of view, the study is well-designed and described. I would only suggest adding recent references to support the studies in lines 285-286. Finally, I´m confused about the cassette for ESC selection in the vector, ampicillin, or neomycin. Lines 82 and 101.
Author Response
Response to Reviewers
We appreciate the Reviewers’ critical review and constructive suggestions for this manuscript. In response to their comments, we have revised the manuscript by re-organizing data, correcting errors in statistics and mouse generation strategy, and adding new data and supportive materials. We believe that the revised manuscript is substantially improved by these revisions. Following is a detailed, point-by-point responses to the Reviewers.
Reviewer 3
The authors characterized a novel TG mouse and show that OE of HADC9 induces adipocyte hypertrophy, but also leads to impaired insulin resistance and hepatic steatosis. If well the authors indicate the limitations of their work, they also recognize that additional studies will be required to link HDAC9 with other metabolic diseases.
On the other hand, from my point of view, the study is well-designed and described. I would only suggest adding recent references to support the studies in lines 285-286.
Authors' response: Thank you for your suggestion. We have added more references shown below.
- Bagchi RA, Weeks KL. Histone deacetylases in cardiovascular and metabolic diseases. J Mol Cell Cardiol. 2019 May;130:151-159.
- Yang C, Croteau S, Hardy P. Histone deacetylase (HDAC) 9: versatile biological functions and emerging roles in human cancer. Cell Oncol (Dordr). 2021 Oct;44(5):997-1017.
Finally, I´m confused about the cassette for ESC selection in the vector, ampicillin, or neomycin. Lines 82 and 101.
Authors' response: We apologize for the mistake. To achieve more consistent transgene expression, PiggyBAC system was used, in which transposons integrate a copy of the transgene into the flanking TTAA sequence, leading to more precise and consistent expression of the target gene. Note that the transgenic mouse line was constructed via a contract with Cyagen Inc. We have extensively modified the description for HDAC9 TG mice generation in Materials and Methods section. We appreciate the Reviewer for pointing this error out.
Round 2
Reviewer 1 Report
Comments and Suggestions for Authors
Most of points were suitably revised in biomolecules-2894379-v2. A few points to be re-considered/changed (pointed out in <Comments to Authors>) for the benefit of readers. These are given below.
<Point>
1. In Abstract, “Hdac9” (lines 18 and 28) should be changed to “HDAC9” because the TG mice express HDAC9. In addition, “HDAC9” (line 464) should also be changed to “Hdac9”.
2. Concerning Figure 1C and D, did the authors use the mouse/human common primers for Hdac9? If so, it should be clearly described.
3. The mouse size seems to be no difference in Figure 2A although body weight of TG mice (Figure 2B) was significantly lower. For the benefit (better understanding) of readers, please replace Figure 2A to another photo in which TG is apparently smaller than WT.
